# New Insights into the Effect of Fipronil on the Soil Bacterial Community

**DOI:** 10.3390/microorganisms11010052

**Published:** 2022-12-23

**Authors:** Suzana Eiko Sato Guima, Francine Piubeli, Maricy Raquel Lindenbah Bonfá, Rodrigo Matheus Pereira

**Affiliations:** 1Department of Biochemistry, Institute of Chemistry, University of São Paulo (USP), Sao Paulo 05508000, Brazil; 2Department of Microbiology and Parasitology, Faculty of Pharmacy, University of Sevilla, 41012 Sevilla, Spain; 3Faculty of Biological and Environmental Sciences, Federal University of Grande Dourados (UFGD), Dourados 79804970, Brazil

**Keywords:** bioremediation, 16S rRNA, next generation sequencing, environmental biotechnology, targeted sequencing

## Abstract

Fipronil is a broad-spectrum insecticide with remarkable efficacy that is widely used to control insect pests around the world. However, its extensive use has led to increasing soil and water contamination. This fact is of concern and makes it necessary to evaluate the risk of undesirable effects on non-target microorganisms, such as the microbial community in water and/or soil. Studies using the metagenomic approach to assess the effects of fipronil on soil microbial communities are scarce. In this context, the present study was conducted to identify microorganisms that can biodegrade fipronil and that could be of great environmental interest. For this purpose, the targeted metabarcoding approach was performed in soil microcosms under two environmental conditions: fipronil exposure and control (without fipronil). After a 35-day soil microcosm period, the 16S ribosomal RNA (rRNA) gene of all samples was sequenced using the ion torrent personal genome machine (PGM) platform. Our study showed the presence of Proteobacteria, Actinobacteria, and Firmicutes in all of the samples; however, the presence of fipronil in the soil samples resulted in a significant increase in the concentration of bacteria from these phyla. The statistical results indicate that some bacterial genera benefited from soil exposure to fipronil, as in the case of bacteria from the genus *Thalassobacillus*, while others were affected, as in the case of bacteria from the genus *Streptomyces*. Overall, the results of this study provide a potential contribution of fipronil-degrading bacteria.

## 1. Introduction

Fipronil is an organic insecticide of the phenylpyrazole family, discovered and developed by Rhône-Poulenc between 1985 and 1987 [1] and marketed since 1993 [1]. Its increasing widespread use has raised concerns about the possible effects on the integrity of humans and animals [1,2,3,4]. Fipronil can have a half-life ranging from 15 to 105 days, depending on the soil type and the application of residues such as vinasse and filter cake. The pH of the aqueous medium can also affect the half-life of more than 100 days, which decreases to 28 days at pH 9 and 2.4 at pH 12 [5]. A cytotoxic effect of fipronil and its metabolites in neuroblastoma-derived cells has been reported, and this effect may lead to neurodegenerative diseases [6,7,8]. It has also been reported that fipronil acts as a neurotransmitter receptor and alters the function of neurons [9,10]. This compound and its metabolites can also cause hypertension and memory impairment, which has been observed in fipronil-treated rats [11]. In addition, there are reports of exposure of liver and kidney cells to fipronil, which can lead to the failure of these organs [12,13,14].

Fipronil can remain in the soil for weeks to months. Half-lives of fipronil in dystroferric Red Latosol (clay texture) soils varied from 18 to 100 days, while in dystrophic Red Latosol (medium texture) soils they ranged from 25 to 141 days, with fipronil concentrations of 5 μg per gram of soil [5]. Another study reported half-lives of fipronil in clay loam soils containing 0.50 mg/kg of the pesticide, which varied from about 4 to 6 days, while the value in the soil (control) was about 100 days [15]. The possible impacts of the use of fipronil on the environment and non-target species [3,4,5,15] and its persistence in soil [5,15] are attracting increasing attention in research and in the search for microorganisms that have the potential to biodegrade this compound, as the use of fipronil is of great interest to the environment. For this reason, knowledge of the microbial community in the soil can provide information that favors the search for microorganisms with biotechnological potential, since this ecosystem can be considered the habitat with the greatest bacterial diversity [16,17]. In this sense, the study of the existing microbial community allows not only a better understanding of the ecological dynamics of the soil but also the possibility to search for and exploit microorganisms with promising applications, such as the degradation of recalcitrant compounds [18,19,20,21].

In this regard, metabarcoding is a powerful approach that allows the characterization of the unknown bacterial community in soil samples without the need to culture them [22]. Selected genetic markers in microbial genomes are used to verify the occurrence in the microbiota of different environments, including soil [23]. Recently, Rawat and Joshi [20] reported that this type of next-generation sequencing (NGS) technique has proven to be a valuable tool for biodiversity analysis in metagenomic samples and metabolic pathway prediction. It has been widely used for the microbial diversity analysis of environmental interest [24,25]. NGS technology tags conserved regions of evolutionary markers to find operational taxonomic units (OTU) and expand the knowledge of complex microbial communities [26].

Regarding pesticides, some may be toxic to microorganisms, while others serve as a source of nutrients and energy and can be used for effective bioremediation. In this context, microorganisms capable of degrading fipronil have been isolated and studied [21,27,28]. However, many studies aiming at finding them for biotechnological applications are limited to their isolation because more than 99% of microorganisms cannot be cultured in media [29,30]. Moreover, the pathway of fipronil degradation is still not fully understood. Therefore, metagenomics and metabarcoding (culture-independent methods) enable the exploration of the genetic material of these non-culturable microorganisms. This research aims to answer the hypothesis of whether the application of fipronil affects microbial diversity in microcosms. For this purpose, we have analyzed the bacterial diversity in soil microcosms under short-term exposure to fipronil using metabarcoding analysis using the 16S ribosomal RNA (rRNA) gene and compared the observed diversity with soils not exposed to fipronil.

## 2. Materials and Methods

### 2.1. Soil Sampling and Microcosm

Semideciduous forest soil was collected at the experimental farm of EMBRAPA Agropecuária Oeste, in the region of Dourados-MS Brazil at three coordinates: (1) 22°17′06.4″ S 54°48′37.5″ W; (2) 22°17′05.7° S 54°48′39.9° W; and (3) 22°17′04.9″ S 54°48′42.8″ W. The soil was obtained from three depth levels: soil surface, about 5 cm, and 10 cm depth from the surface. All soil samples were homogenized. Information on the chemical properties of the soil can be found in Table 1.

The experimental design consisted of three samples with the application of fipronil (F1, F2, and F3) and three samples for the control (C1, C2, and C3). The fipronil used was Regent 800 produced and sold by BASF, with a purity of 98.8%. In each autoclaved Erlenmeyer of 250 mL, 30g of the homogenized soil was distributed followed by 6 mL of 0.9% saline solution. All samples were kept at 30 °C in the biochemical oxygen demand incubator during the microcosm period without agitation. In the second week, 3 mL of 0.9% saline was applied to each sample. In the fipronil treatment (F1, F2, and F3 samples), sterilized (by autoclaving) and diluted fipronil was added to the samples with a final concentration of 200 μg soil Kg^−1^. In the third and fourth weeks, 1 mL of 0.9% saline solution was applied to all samples, and in the fipronil treatment, fipronil was added the same way as in the second week (methodology adapted from Silva et al. [31]). The most relevant steps used in this work are described in detail in Figure 1.

### 2.2. DNA Extraction and Metagenomic Library Construction

After four weeks of the microcosm, DNA from the soil samples was extracted using the MoBio Laboratories Power Soil^®^ DNA Isolation kit according to the manufacturer’s instructions. The 16S rRNA metagenomic library construction and the sequencing of DNA amplicons were performed in the central multi-user laboratory for large-scale DNA sequencing (LMSeq) (Facility FAPESP proc. No. 2009/53984-2), at the State University of São Paulo’s (UNESP) Jaboticabal Campus in São Paulo, Brazil.

The extracted DNA was quantified using the Qubit^®^ Fluorometer (Invitrogen-Life Technologies, Waltham, MA, USA). For the metagenomic library, primers for the V4 and V5 regions of the 16S rRNA gene were used for PCR amplification (515F 5′-TGTGNCAGCMGCCGCGGTAA-3 and 926R 5′-barcode-CCCCGYCAATTYMTT-3′) [32]. The primers for amplification were connected to the sequencing adapters, whereas the reverse primer had both the adapter and barcode for sequence identification. The conditions for 16S rRNA amplification were: initial denaturation at 95 °C for 3 min; 35 three-step cycles: 95 °C for 30 s, 55 °C for 1 min and 30 s, and 72 °C for 45 s; and a final extension at 72 °C for 5 min. Agarose gel electrophoresis was performed for the amplicons. The bands formed corresponding to the V4 and V5 regions of 16S rRNA were cut from the gel and purified with the Zymoclean™ Gel DNA Recovery kit according to the manufacturer’s instructions (Zymo Research, The Epigenetics Company, Irvine, CA, USA).

### 2.3. High-Throughput Sequencing

The amplified sequences of the V4 and V5 regions of the 16S rRNA gene were quantified using the Qubit^®^ Fluorometer. The mean size of the fragments was verified using the 2100 Bioanalyzer (Agilent, Santa Clara, CA, USA). The quantity and size data of the DNA sequences were used for the dilution and emulsion preparation of the samples.

The Ion OneTouch™ 2 system and the Ion PGM Hi-Q OT2 400 kit were used for the emulsion and enrichment of the samples. DNA was deposited on a 318 V2 Ion chip, and sequencing was conducted on a PGM Ion-Torrent Hi-Q following the manufacturer’s instructions.

Raw reads with a minimum size of 150 bp and a minimum quality mean Q20 were filtered through the Prinseq software v.0.20.4 [33]. Cutadapt v.1.14 [34] was used to trim primers and adapters, deleting sequences in which no adapter was found. Sequences shorter than 150 bp were also discarded.

Raw data from 16S rRNA bacterial was deposited into the Sequence Read Archive (SRA) database of the National Center for Biotechnology Information (NCBI) and can be accessed by BioProject ID PRJNA692250 (https://www.ncbi.nlm.nih.gov/bioproject/PRJNA692250, accessed on 14 January 2021).

### 2.4. Diversity Analysis

The Brazilian Microbiome Project (BMP) pipeline [35] was performed with adaptations. For the following steps, USEARCH v.9 [36] was used. Sequences were deduplicated. Their abundances were classified from highest to lowest. The singletons found (OTUs represented by only a single DNA sequence) were discarded. In order to characterize the taxonomic structure of the samples, sequences were grouped in operational taxonomic units (OTUs) using the UPARSE method [37] and considering the 97% identity, which represents the usual definition for bacteria genus. OTUs were mapped to the initial file after preprocessing. The resulting file from the mapping step was converted into tabulated text format. The Biological Observation Matrix (BIOM) script [38] was then used to convert the text file into BIOM format (OTUs table) and to add metadata to the file.

Quantitative Insights into Microbial Ecology (QIIME) v.1.9.1 [39] was used for the following steps. Taxonomy was attributed to clustered and filtered OTUs using the Ribosomal Database Project (RDP) method [40]. Global alignment of the filtered OTUs was performed using the Greengenes reference alignment core [41]. As a standard, QIIME executes the alignment using the Python Nearest Alignment Space Termination (PyNAST) program method [42]. Alignments were filtered using QIIME. A reference tree was constructed from the filtered global alignment with the set of representative OTU sequences using the FastTree program [43] (QIIME default software for alignments).

Multiple rarefaction curves were constructed using QIIME (multiple_rarefactions.py script) from the BIOM table, with step sizes of 100 sequences, iteration of 50, and a maximum depth of 11,547 (the number of sequences of the smallest sample). From the multiple rarefaction files and the reference tree, the number of OTUs, Shannon, Simpson, Chao1, and Phylogenetic Diversity (PD) Whole Tree indexes were calculated using the alpha_diversity.py script from QIIME. A rarefaction curve was constructed for the number of OTUs and each alpha-diversity index through the collate_alpha.py and make_rarefaction_plots.py scripts. Phylum proportion graphics for each sample were generated using MEGAN v.6.6.7 [44] by importing the BIOM file.

### 2.5. Statistical Analysis

The BIOM table file was converted into a tabulated format containing the number of sequences for each OTU on each sample and imported into R. Then, the Levene test (from the car package) [45] and the Shapiro-Wilk test (from the stats package) [46] were performed on the table to check, respectively, whether the sample data had equal variance and normal distribution.

BIOM table was converted into a STAMP Profile File (spf) format using the biom_to_stamp.py script from the Microbiome Helper workflow [47]. Statistical Analysis of Taxonomic and Functional Profiles (STAMP) v.2.1.3 [48] was used to construct the principal component analysis (PCA) plot based on the genus level. Because the sample data had heteroscedasticity and non-normal distribution, as demonstrated by the Levene test and Shapiro–Wilk test (considering *p*-value < 0.05 for both tests), the nonparametric White *t*-test [49] was selected on the STAMP software to infer the differences in mean proportions between both groups. The selected method for the confidence interval was DP: bootstrap of 0.95.

## 3. Results

### 3.1. General Information about Sampling and Sequencing

As a result of sequencing all the samples studied, a total of 321,561 raw reads were obtained. These reads were preprocessed using the Prinseq program, resulting in 295,258 reads, of which 204,855 were trimmed with cutadapt and had final length longer than 150 bp. The resulting N50 (the size of the reads after the cutting process) was 341 bp. After clustering, a total of 271 OTUs were obtained.

Ecological indices were used to verify whether there was a variation in diversity and species richness between the different treatments analyzed. In addition to the number of OTUs, Shannon and Simpson indices were calculated and are represented in rarefaction curves (Figure 2). All the generated rarefaction curves reached the stabilization shown by the plateau (asymptomatic values on the Y-axis).

In Figure 2A, the use of the Shannon–Weaver ecological index revealed a high species richness. Interestingly, although there is a high species richness, the results do not indicate a significant difference between the fipronil-exposed and control samples.

Simpson’s reciprocity index, in turn, refers to the probability that two random species in the samples belong to the same species. The higher the value, the higher this probability. Figure 2B shows a high value of species diversity but no difference between the control and fipronil-exposed samples.

Finally, Figure 2C shows that the sequencing depth was sufficient to capture all the microbial diversity in the analyzed sample. This is indicated by the plate in the rarefaction curve obtained from the dataset.

In the PCA plot (Figure 3), fipronil and control microcosm soil presented different taxonomic profiles. Taking into account the highest axis (PC1) in the PCA, the control samples had a shorter distance between themselves compared to the distance between the samples within the fipronil group. The difference in PCA grouping between control and fipronil samples indicates that fipronil affected the evaluated bacterial communities.

### 3.2. Composition of the Microbial Community

The diversity of the microbial community found in the samples was analyzed and represented in Figure 4. Eleven different bacterial phyla were identified, with Proteobacteria, Actinobacteria, and Firmicutes predominating in all samples.

Consistently among each sample, the mean proportion of the most abundant phyla for the control and fipronil-exposed samples, respectively, were Proteobacteria 33.12 ± 5.51% and 35.69 ± 2.79%; Actinobacteria 23.59 ± 3.40% and 26.73 ± 3.62%; and Firmicutes 17.52 ± 5.82% and 25.02 ± 11.04%.

Finally, the range of OTUs was compiled and ordered by summing the readings of all samples for each OTU (Table 1). These results revealed that of all the bacterial genera observed in Table 2, *Thalassobacillus*, a member from the Firmicutes phylum, showed the greatest difference between the two conditions studied.

### 3.3. Statistically Significant Changes

Based on the Levene and Shapiro-Wilk tests, the distribution of the data did not conform to the assumption of homogeneity (F statistics = 3.3198; *p*-value = 0.005482) nor normality (W = 0.31539; *p*-value < 2.2 × 10^−16^). Since the nonparametric White *t*-test eliminates the normality assumption of a standard *t*-test by using permutation, this statistical hypothesis test was selected to infer the differences between both groups studied in this work. At the phylum level, there were no statistically significant differences between the treatments, considering the *p*-value to be less than 0.05. At the genus level, differences occurred for six genera (Figure 5). Interestingly, *Thalassobacillus* and *Streptomyces* were the genera with the highest effect sizes for fipronil and control, respectively. The populations from the genus *Thalassobacillus* were higher in the fipronil-treated samples, while those from the genus *Streptomyces* were higher in the control samples. The unclassified and uncultured genera were omitted from the analysis.

## 4. Discussion

To validate the representativeness of the samples, rarefaction curves were constructed. The stabilization of rarefaction curves for Shannon and Simpson indices and OTU number presented sufficient sequence coverage for taxonomic diversity. Similarly, the rarefaction curves for Chao1 and PD whole tree also tended to *plateau* (Figure 2). This behavior indicates that the higher the number of sequences considered in the samples, the lower the contribution of these additional sequences to the diversity index and the number of OTUs. Thus, the sample size was considered adequate to illustrate the diversity of the soil microcosm bacterial community.

Soil samples were collected in a transition region between the Cerrado and the Atlantic Forest. In studies carried out in similar environments, a predominance of the phyla represented by Proteobacteria, Actinobacteria, and Firmicutes has been observed, with approximately 80% of the total number of representatives representing the relative abundance of these three phyla [50,51]. Different studies have shown the occurrence of these same phyla in the Brazilian Cerrado [52,53,54].

In this study, at the phylum level, it was observed that the exposure of the soil samples to fipronil affected the bacterial community with a slight increase in the representatives from the phyla Proteobacteria, Actinobacteria, and Firmicutes, compared to the control samples. Such variation was recently mentioned by Walder et al. [55], who similarly observed that contamination of agricultural soils with pesticide residue influences the microbiome. In this regard, some of the genera observed, such as *Thalassobacillus* (Figure 5) and *Rhodoplanes* (Table 1), have been shown to be positively influenced by fipronil exposure. On the other hand, fewer genera were observed in fipronil-exposed samples compared to the control, suggesting that these genera were influenced negatively by the presence of this compound (Figure 4).

The changes observed in a microbial community may be related to the ability of some microorganisms to degrade recalcitrant compounds. Some work has shown that after exposure to pesticides, some groups of bacteria were able to return to the initial population using the applied pesticides as a source of energy and carbon [56]. In other words, some microbial cultures are able to degrade xenobiotic compounds, such as fipronil, as the sole carbon and energy source. For example, in the recent work published by Singh et al. [20], the authors mentioned that only nine bacteria have been identified as fipronil degraders up to their publication date. Among them, we can mention *Bacillus thuringiensis* [57] and *Streptomyces rochei* [58]. In the same vein, Cappelini et al. [59] observed that the bacterium *Burkholderia thailandensis* belonging to the phylum Proteobacteria also showed potential for fipronil degradation. Another example is the result observed by Ahemad et al. [60], where the authors stated that the *Rhizobium* strain MRL3 could grow in a medium with fipronil as the sole source of carbon and nitrogen. Finally, Gangola et al. [61] have demonstrated the potential of *Bacillus* sp. strain 3C to degrade environmental pesticide mixtures. In this work, changes in the microbial community have been observed, with the different taxonomic profiles demonstrated by PCA (Figure 3) indicating that the bacterial community composition is different between the control and fipronil soil microcosms. Additionally, the positive influence of fipronil exposure on the genera *Thalassobacillus* and *Rhodoplanes* could suggest the involvement of these two groups in the fipronil degradation pathway.

In terms of bacterial community composition, Proteobacteria was the most predominant phylum in all samples. The same was observed in other metagenomic studies on soil microbiomes from the Cerrado [51,52,53,54] and the Atlantic Forest [61]. In this study, as mentioned above, the phyla Proteobacteria, Actinobacteria, and Firmicutes were observed in all samples. However, soil samples exposed to fipronil showed a higher population of bacteria representative of these phyla. In Table 2, two bacteria (*Gaiella* and *Actinobacterium* Gp-6) are among the ten most abundant OTUs for each soil microcosm sample. Both bacteria belong to the phylum Actinobacteria and experienced a slight increase after fipronil application. In line with the results obtained in this work, Pacchioni et al. [62] have described bacteria from the phylum Actinobacteria in fipronil degradation experiments in soil. The authors also stated that the same strain increased the exopolysaccharide secretion with progressively increasing fipronil concentrations. Similarly, Tomazini et al. [63] recently showed that the strain *Mesorhizobium* sp. MRC4, from the same order as *Rhizobium*, was tolerant to high levels of several pesticides. The authors also observed that this strain produced increased exopolysaccharide secretion with increasing pesticide concentrations. In another recent study, Araujo et al. [54] observed that different strains of *Mesorhizobium* were shown to be highly tolerant to different pesticides used in chickpea cultivation. In this context, the results presented in Table 2 show the presence of *Mesorhizobium* in higher concentrations in soil samples exposed to fipronil, confirming the observations of the studies cited above. In addition, most genera from the order *Mesorhizobium* are beneficial to their plant hosts. They are often associated with nitrogen fixation, nodulation of legumes, and symbiosis with plant roots [64].

With respect to Actinobacteria, this phylum was also one of the most abundant in all samples. Among the genera belonging to Actinobacteria, *Streptomyces* was the most affected by the exposure of soil samples to fipronil, with a decrease in its population in the presence of this compound. Consistent with the high abundance in the samples, *Streptomyces* is commonly found in soils [65]. With respect to that genus, members of this group of microorganisms include important wood decomposers and producers of secondary metabolites, such as antibiotics [66,67]. These compounds influence their antagonism against other microorganisms. They are also capable of synthesizing proteases, chitinases, and xylanases [67].

In addition to the most abundant phyla, representatives from the Firmicutes group were present in all samples. Interestingly, some bacteria from the phylum Firmicutes have been described as involved in fipronil degradation [62]. In this work, the phylum includes *Bacillus* and *Thalassobacillus* among the ten most abundant OTU in the samples. Representatives from this phylum are spore-forming microorganisms, which increases their chances of survival in disturbed environments. Although no statistical differences between treatments were found for *Bacillus* in this study, research has reported fipronil biodegradation promoted by different *Bacillus* species [15,27,57,68,69]. Interestingly, *Thalassobacillus*, a genus from the same family as *Bacillus*, had a higher proportion of representatives in the fipronil-exposed microcosms compared to the control. In relation to this group of microorganisms, it should be mentioned that the bacterium *Thalassobacillus devorans* was isolated for the first time from saline soils in southern Spain [70]. Many other species belonging to the genus *Thalassobacillus* have been isolated from various saline environments, such as a hypersaline lake, marine coastal tidal flat sediments, and salty animal skins [71,72,73,74]. *Thalassobacillus* bacteria are moderately halophilic. Consequently, the saline solution applied to the microcosms could have been conducive to their growth. Considering that *Thalassobacillus* can biodegrade aromatic compounds, the difference in ratios between fipronil and the control suggests that *Thalassobacillus* may be able to metabolize fipronil or fipronil-derived compounds. If this is correct, it is possible that *Thalassobacillus* may contribute to the biodegradation of the pesticide.

Finally, by using 16S rRNA metagenomics, this study indicates changes in taxonomic groups due to fipronil exposure. These findings motivate future research on the experimental effects of fipronil on specific bacterial groups, especially the candidates with the potential to degrade the fipronil described here. Using this information as a basis for the development of new tools, efforts can be made to minimize the impacts of human activity on soil microbial activities and biochemical cycling and to increase crop productivity by facilitating the abundance of plant growth-promoting bacteria.

## 5. Conclusions

This work showed that exposure of soil samples to fipronil affects the bacterial community in studies using microcosms. Therefore, the hypothesis proposed was confirmed. In this sense, it has been observed that the phyla represented by Proteobacteria, Actinobacteria, and Firmicutes have benefited from soil exposure to fipronil, with their population slightly higher in the presence of fipronil compared to the control.

The findings of this study point in the direction of the potential bacterial candidates for fipronil degradation, providing valuable information for further research. The metabolic pathway for fipronil and its metabolite degradation is still not completely understood, but the effects of its indiscriminate use have been demonstrated on some organisms, emphasizing the importance of investigating ways to degrade the pesticide. Collaborative efforts to better understand the relationship between the microorganisms and their contact with the pesticide can shed light on the metabolic pathway for fipronil and its metabolite degradation in the future, minimizing the negative impacts of human intervention on the environment. 

## Figures and Tables

**Figure 1 microorganisms-11-00052-f001:**
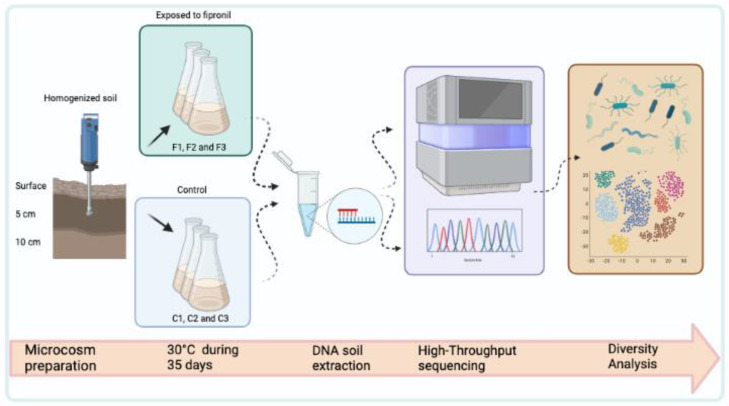
Schematic representation of the main steps carried out in this work.

**Figure 2 microorganisms-11-00052-f002:**
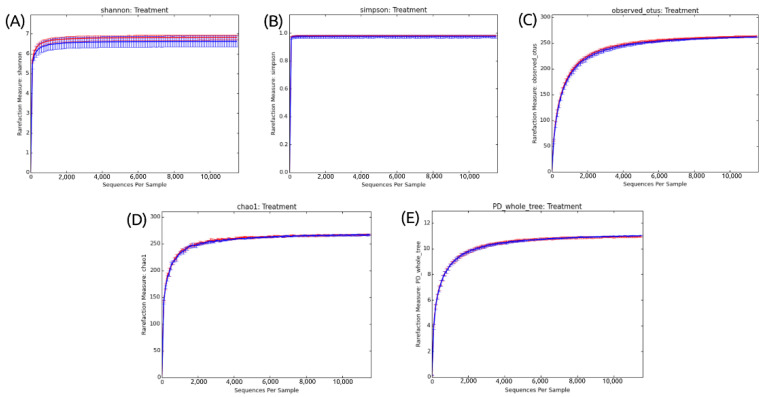
Alpha diversity metric indexes and number of OTUs in the fipronil and control groups. (**A**) Stabilization of the rarefaction curves for the Shannon index. (**B**) Stabilization of the rarefaction curves for the Simpson index. (**C**) Stabilization of the rarefaction curves for the number of OTUs. (**D**) Stabilization of the rarefaction curves for the Chao1 index. (**E**) Stabilization of the rarefaction curves for the Phylogenetic Diversity Whole Tree index. Rarefaction curve generated using QIIME. The rarefaction curves used a step size of 100. The control is plotted in red, and the fipronil is plotted in blue.

**Figure 3 microorganisms-11-00052-f003:**
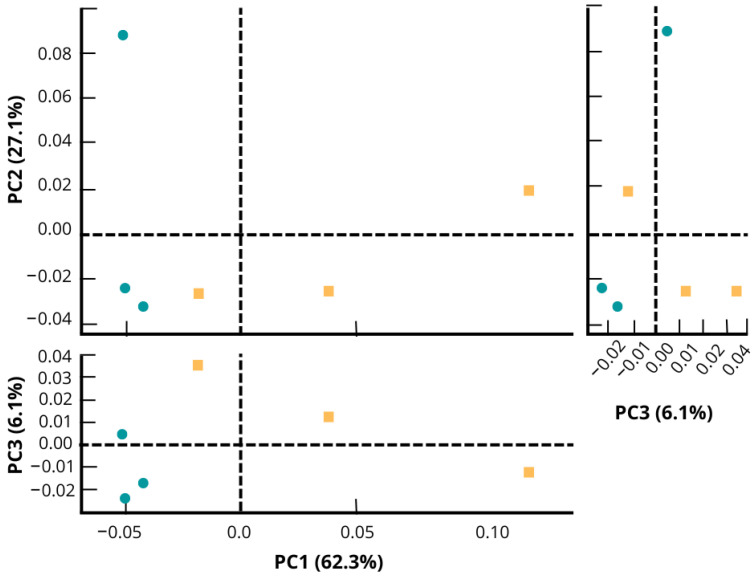
Principal component analysis representing the structure of the bacterial community of soil microcosm under application of fipronil and control at the genus level. The PCA was generated using the software STAMP. Control samples are plotted in green color and fipronil in yellow color. Each axis shows the variance explained in percentage of the distance between the bacterial community structures of the samples.

**Figure 4 microorganisms-11-00052-f004:**
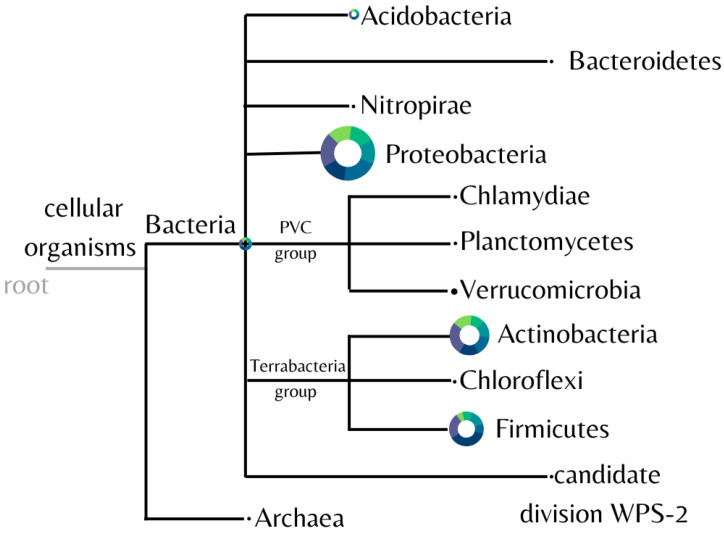
Identified phyla from soil samples. Image generated by the software MEGAN using the biom file from the BMP pipeline. The control is plotted in green, and the fipronil is plotted in blue. The size of the circle is linearly proportional to the absolute abundance found for each phylum summing all the samples.

**Figure 5 microorganisms-11-00052-f005:**
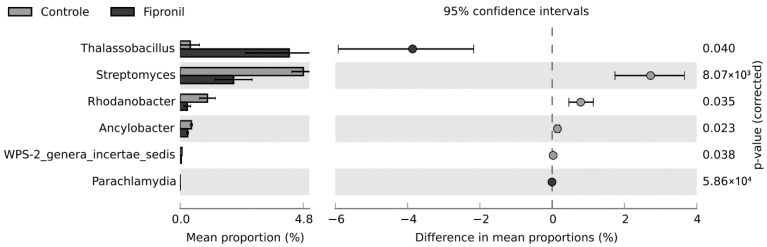
Extended bar for the difference in mean proportions between fipronil and control treatments at the genus level. Nonparametric White *t*-test was used with confidence interval DP: boostrap 0.95. Only results with *p*-value less than 0.05 are shown.

**Table 1 microorganisms-11-00052-t001:** Read chemical analysis of the soil collected.

Chemical Analysis	Value
P (mg/dm^3^)	3.46
Ca (cmol_c_/dm^3^)	4.04
Mg (cmol_c_/dm^3^)	1.93
K (cmol_c_/dm^3^)	0.27
Al (cmol_c_/dm^3^)	0.00
H + Al (cmol_c_/dm^3^)	2.30
pH CaCl_2_	6.19
pH H_2_O	6.74
pH SMP	6.83
T (cmol_c_/dm^3^)	8.52
SB (cmol_c_/dm^3^)	6.23
V%	73.06

**Table 2 microorganisms-11-00052-t002:** Read counts of the top ten most abundant OTUs for each soil microcosm sample. Control samples: C1, C2, and C3 and Fipronil samples: F1, F2, and F3. Norm is normalization (the mean relative abundance).

Genera	C1	C2	C3	Sum	Norm.	F1	F2	F3	Sum	Norm.
*Gaiella*	359	356	514	1229	0.057219	919	621	604	2144	0.099818
*Thalassobacillus*	0	102	38	140	0.006518	839	1632	480	2951	0.13739
*Mesorhizobium*	360	301	273	934	0.043484	693	485	623	1801	0.083849
*Bacillus*	1211	4	10	1225	0.057032	15	1214	15	1244	0.057917
*Streptomyces*	502	377	490	1369	0.063737	315	244	417	976	0.04544
*Mycobacterium*	252	191	255	698	0.032497	423	345	308	1076	0.050095
*Actinobacterium Gp-6*	236	201	247	684	0.031845	347	377	276	1000	0.046557
*Bradyrhizobium*	199	185	186	570	0.026538	379	228	329	936	0.043577
*Pedomicrobium*	184	161	164	509	0.023698	303	271	283	857	0.039899
*Rhodoplanes*	138	134	153	425	0.019787	237	217	257	711	0.033102

## Data Availability

The raw data of bacterial 16S rRNA was deposited into Sequence Read Archive (SRA) database on the Nacional Center for Biotechnology Information (NCBI) and can be accessed by BioProject ID PRJNA692250. The SRA database is public and can be freely accessed through the NCBI website.

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
