# Peer review of "New Insights into the Effect of Fipronil on the Soil Bacterial Community"

_microorganisms, 2022, doi:10.3390/microorganisms11010052_

Round 1
Reviewer 1 Report
Review Report of Manuscript No. microorganisms-2098512
In the manuscript entitled “New insights into the effect of fipronil on the soil bacterial community”, authors have investigated the effect of fipronil on the soil bacterial community. The increasing use of the insecticide fipronil has led to widespread soil and water contamination. Thus, it requires attention to assessing the risk of undesirable effects on nontarget microorganisms, such as the microbial community present in water and/or soil. The topic selection of the manuscript is novel, in line with the current hot spots in the environmental field. In totality, this paper is pleasant to read, well-structured and well-written. However, it needs some corrections and there are some queries which the authors should kindly respond to make it good.
Some specific suggestions or questions are listed below:
1. Introduction is easy to read but needs a little completed. For example, what is the half-life of fipronil in the environment? What is the specific concentration of fipronil detected in the environment? I think it is favorable to show this information in this part based on the literature available and described in original paper. This way the authors will demonstrate that they really have a good knowledge of the related literature.
2. Where is the pesticide fipronil obtained? And the purity? Please add this information in the Materials and Methods section.
3. Line 126: 2.2.DNA Extraction and Metagenomic Library Construction. Please use DNA or RNA uniformly throughout the manuscript. Check the abstract, line 87 (16S rRNA), and line 375 (16S rRNA).
4. Table 1, there are some overlaps in the table , please revise it.
5. Line 162: NCBI, please add the website here.
6. Statistical analysis is very important. I suggest the authors add a new section in the Materials and Methods to describe the details of the statistical analysis.
7. Figure 4 can be improved.
8. Line 300: 4. Discussion. This section can be improved. Recently, several papers have investigated the toxicity and biodegradation of fipronil by soil bacterial communities in contaminated environment such as Novel mechanism and degradation kinetics of pesticides mixture using Bacillus sp. strain 3C in contaminated sites. Pestic Biochem Physiol. 2022, 181:104996; A comprehensive review of environmental fate and degradation of fipronil and its toxic metabolites. Environ Res. 2021,199:111316; Insights into the toxicity and biodegradation of fipronil in contaminated environment. Microbiological Research, 2023, 266: 127247. The authors should add more information into this section and cite the recent research into the field.
9. Conclusions: This section can be improved and written more main conclusions. In addition, authors can add and revise this section for the better understanding of the topic and its future research.
10. There are many abbreviations/acronyms in the manuscript. Please check throughout the manuscript that all abbreviations/acronyms are defined the first time they appear in each of three sections: the abstract; the main text; the first figure or table.
11. Please check all the species names. Species names are typically given in full the first time they are used within the main text and then abbreviated throughout the remainder of the text.
12. References: Many of the references have been superceded and more modern ones are required such as Tingle, C. C. D.; Rother, J. A.; Dewhurst, C. F.; Lauer, S.; King, W. J. (2003); Stevenson, B. S.; Eichorst, S. A.; Wertz, J. T.; Schmidt, T. M.; Breznak, J. A (2004); García, M. T.; Gallego, V.; Ventosa, A.; Mellado, E (2005).
Author Response
Response to Reviewer 1 Comments
In the manuscript entitled “New insights into the effect of fipronil on the soil bacterial community”, authors have investigated the effect of fipronil on the soil bacterial community. The increasing use of the insecticide fipronil has led to widespread soil and water contamination. Thus, it requires attention to assessing the risk of undesirable effects on nontarget microorganisms, such as the microbial community present in water and/or soil. The topic selection of the manuscript is novel, in line with the current hot spots in the environmental field. In totality, this paper is pleasant to read, well-structured and well-written. However, it needs some corrections and there are some queries which the authors should kindly respond to make it good.
Some specific suggestions or questions are listed below:
Point 1. Introduction is easy to read but needs a little completed. For example, what is the half-life of fipronil in the environment? What is the specific concentration of fipronil detected in the environment? I think it is favorable to show this information in this part based on the literature available and described in original paper. This way the authors will demonstrate that they really have a good knowledge of the related literature.
Answer: Thank you for providing this suggestion, we have added the information to the manuscript
Point 2. Where is the pesticide fipronil obtained? And the purity? Please add this information in the Materials and Methods section.
Answer: We agree with you and have incorporated this suggestion throughout our paper. The fipronil used was Regent 800 produced and sold by BASF, purity is 98.9%.
Point 3. Line 126: 2.2.DNA Extraction and Metagenomic Library Construction. Please use DNA or RNA uniformly throughout the manuscript. Check the abstract, line 87 (16S rRNA), and line 375 (16S rRNA).
Answer. We have replaced the term 16S throughout the paper with 16S rRNA to use more precise terms.
Point 4. Table 1, there are some overlaps in the table, please revise it.
Answer. We have reviewed the table
Point 5. Line 162: NCBI, please add the website here.
Answer. We agree with you and have inserted the website address of the SRA database and the NCBI in the article.
Point 6. Statistical analysis is very important. I suggest the authors add a new section in the Materials and Methods to describe the details of the statistical analysis.
Answer. Thank you for providing these insights. We agree with you and have incorporated this suggestion throughout our paper.
Point 7. Figure 4 can be improved.
Answer. We have included a new Figure 4 to further illustrate the Identified of the phylas from soil samples.
Point 8. Line 300: 4. Discussion. This section can be improved. Recently, several papers have investigated the toxicity and biodegradation of fipronil by soil bacterial communities in contaminated environment such as Novel mechanism and degradation kinetics of pesticides mixture using Bacillus sp. strain 3C in contaminated sites. Pestic Biochem Physiol. 2022, 181:104996; A comprehensive review of environmental fate and degradation of fipronil and its toxic metabolites. Environ Res. 2021,199:111316; Insights into the toxicity and biodegradation of fipronil in contaminated environment. Microbiological Research, 2023, 266: 127247. The authors should add more information into this section and cite the recent research into the field.
Answer. Thank you for your suggestion. The discussion has been improved and the items mentioned have been included.
Point 9. Conclusions: This section can be improved and written more main conclusions. In addition, authors can add and revise this section for the better understanding of the topic and its future research.
Answer. The conclusions have been rewritten and we have added the perspectives for the future.
Point 10. There are many abbreviations/acronyms in the manuscript. Please check throughout the manuscript that all abbreviations/acronyms are defined the first time they appear in each of three sections: the abstract; the main text; the first figure or table.
Answer. Thank you for your observation. All abbreviation and/ or acronyms were revised
Point 11. Please check all the species names. Species names are typically given in full the first time they are used within the main text and then abbreviated throughout the remainder of the text.
Answer. All the species names were revised
Point 12. References: Many of the references have been superceded and more modern ones are required such as Tingle, C. C. D.; Rother, J. A.; Dewhurst, C. F.; Lauer, S.; King, W. J. (2003); Stevenson, B. S.; Eichorst, S. A.; Wertz, J. T.; Schmidt, T. M.; Breznak, J. A (2004); García, M. T.; Gallego, V.; Ventosa, A.; Mellado, E (2005).
Answer. We agree with you and these references have been replaced by more recent ones.
Reviewer 2 Report
The manuscript entitled "New insights into the effect of fipronil on the soil bacterial community" has been reviewed
1) Please thoroughly check and fix the manuscript for typos and grammatical errors.
2) Please add the hypnosis at the end of the introduction.
3) Please put the country (Brazil) after in the region of Dourados.
4) Changes all numbers in your manuscript from this format 3,46 to 3.46
5) Based on what evidence was this concentration of fipronil (200 μg ∙ soil Kg-1) used?
6) Authors mentioned that they collected samples from three different sites. Do you sure that these sites are not contaminated with other pesticides? Maybe the effect of the microbial community is due to any other pesticide. Do you make any analysis to confirm these samples are free of pesticide residues?
7) write the full name of BOD incubator when mentioned in the first time in manuscript.
8) I have a confused need author showed it to me, Authors mentioned there used autoclaving to sterilize the soil. Why? When authors use the autoclave, the soil become free of any bacterial.
9) Authors need to confirm 16S rRNA ribosomal subunit gene by make a bioremediation experiment for one of Streptomyces population and Thalassobacillus under greenhouses conditions to show the fipronil degradation in different periods by HPLC or GC/Ms.
10) Discussion sections need more citations.
11) Rewrite the conclusion again.
I am stopping here and after making all modifications mentioned above in the manuscript. I will revise it from A to Z.
Author Response
Response to Reviewer 2 Comments
The manuscript entitled "New insights into the effect of fipronil on the soil bacterial community" has been reviewed
Point 1. Please thoroughly check and fix the manuscript for typos and grammatical errors.
Answer. Thanks for the suggestion. We agree with you and have carried out an extensive revision of the English language.
Point 2. Please add the hypnosis at the end of the introduction.
Answer. Thank you for providing these insights. The hypothesis was introduced in the article.
Point 3. Please put the country (Brazil) after in the region of Dourados.
Answer. We agree with you and have incorporated this suggestion throughout our paper.
Point 4. Changes all numbers in your manuscript from this format 3,46 to 3.46
Answer. All numbers were updated to the recommended standard.
Point 5. Based on what evidence was this concentration of fipronil (200 μg ∙ soil Kg-1) used?
Answer. It was a protocol adapted from the work of SILVA, C. M. M. S.; ROQUE, M. R. A.; MELO, I. S. ed. Environmental microbiology: a laboratory manual. Jaguariúna: EMBRAPA Meio Ambiente, p. 98, 2000.
Point 6. Authors mentioned that they collected samples from three different sites. Do you sure that these sites are not contaminated with other pesticides? Maybe the effect of the microbial community is due to any other pesticide. Do you make any analysis to confirm these samples are free of pesticide residues?
Answer. The semi-deciduous forest soil was collected at the experimental farm. This soil was chosen precisely to avoid possible contamination, as it is a closed area and cared for by the Embrapa company, which never grew anything on the premises, but it should be mentioned that no prior tests were carried out to check for any kind of contamination.
Point 7. write the full name of BOD incubator when mentioned in the first time in manuscript.
Answer. We agree with you and have incorporated this suggestion throughout our paper.
Point 8. I have a confused need author showed it to me, Authors mentioned there used autoclaving to sterilize the soil. Why? When authors use the autoclave, the soil become free of any bacterial.
Answer. The empty Erlenmeyer flask and the fipronil were autoclaved, not the soil. The expression has been modified in the text.
Point 9. Authors need to confirm 16S rRNA ribosomal subunit gene by make a bioremediation experiment for one of Streptomyces population and Thalassobacillus under greenhouses conditions to show the fipronil degradation in different periods by HPLC or GC/Ms.
Answer. You have raised an important point; however, we believe that such confirmation would be beyond the scope of our article because the approach used is more metagenomic, the main objective is to verify whether there is a variation in the microbial population in the presence of fipronil.
Confirming whether the Streptomyces and Thalassobacillus populations are fipronil degraders would be a step further, i.e., in this work we have found a potential value for these microorganisms and we open an excellent precedent for a future search for a real value for them, since this compound is very important in the environmental field and the results observed in this work could justify the use of new efforts to find solutions to environmental pollution by pesticides.
Point 10. Discussion sections need more citations.
Answer. Thank you for your suggestion. We agree with you and the debate has been improved and new quotes have been added.
Point 11. Rewrite the conclusion again.
Answer. The conclusion was rewrite.
I am stopping here and after making all modifications mentioned above in the manuscript. I will revise it from A to Z.
Thank you for your time and for giving us the opportunity to strengthen our manuscript with your valuable comments and queries. We have worked hard to incorporate your feedback and hope that these revisions persuade you to accept our submission.
Round 2
Reviewer 1 Report
The authors have considered most comments raised by the reviewers and revised the manuscript accordingly based on these comments. However, I suggest the authors add two recent publications (A comprehensive review of environmental fate and degradation of fipronil and its toxic metabolites. Environ Res. 2021,199:111316; Insights into the toxicity and biodegradation of fipronil in contaminated environment. Microbiological Research, 2023, 266: 127247) about the topic into the revised manuscript and develop an interesting discussion.
Reviewer 2 Report
Accept